# Effect of Plant Extracts Combinations on TNF-α, IL-6 and IL-10 Levels in Serum of Rats Exposed to Acute and Chronic Stress

**DOI:** 10.3390/plants12173049

**Published:** 2023-08-24

**Authors:** Ilin Kandilarov, Petya Gardjeva, Maria Georgieva-Kotetarova, Hristina Zlatanova, Natalia Vilmosh, Ivanka Kostadinova, Mariana Katsarova, Kiril Atliev, Stela Dimitrova

**Affiliations:** 1Department of Pharmacology and Clinical Pharmacology, Faculty of Medicine, Medical University of Plovdiv, 15A Vassil Aprilov, 4002 Plovdiv, Bulgaria; ilin.kandilarov@mu-plovdiv.bg (I.K.); maria.georgieva@mu-plovdiv.bg (M.G.-K.); hristina.zlatanova@mu-plovdiv.bg (H.Z.); nataliya.vilmosh@mu-plovdiv.bg (N.V.); ivanka.kostadinova@mu-plovdiv.bg (I.K.); 2Department of Medical Microbiology and Immunology „Prof. Dr. Elissay Yanev”, Faculty of Pharmacy, Medical University of Plovdiv, 15A Vassil Aprilov, 4002 Plovdiv, Bulgaria; petya.gardzheva@mu-plovdiv.bg; 3Department of Bioorganic Chemistry, Faculty of Pharmacy, Medical University of Plovdiv, 15A Vassil Aprilov, 4002 Plovdiv, Bulgaria; mariana.katsarova@mu-plovdiv.bg; 4Department of Urology and General Medicine, Faculty of Medicine, Medical University of Plovdiv, 15A Vassil Aprilov, 4002 Plovdiv, Bulgaria; kiril.atliev@mu-plovdiv.bg; 5Research Institute, Medical University of Plovdiv, 15A Vassil Aprilov, 4002 Plovdiv, Bulgaria

**Keywords:** flavonoids, combined plant extracts, TNF-α, IL-6, IL-10, stress and depression

## Abstract

Oxydative stress, anxiety and depression are associated with changes in cytokine levels. Natural products, including individual and combined plant extracts, have the potential to be used in the treatment of neuropsychiatric disorders. The goal of this study is to investigate the effects of two combined plant extracts, rich in flavonoids, on the levels of the cytokines TNF-α, IL-6, and IL-10 in rats subjected to models of acute cold stress and chronic unpredictable stress. The study utilized common medicinal plants such as *Valeriana officinalis*, *Melissa officinalis*, *Crataegus monogyna, Hypericum perforatum*, and *Serratula coronata*, which were combined in two unique combinations—Antistress I and Antistress II. The compositions of the used extracts were determined by HPLC methods. Pro- and anti-inflammatory cytokines in rats’ serum were measured with Enzyme-linked immunosorbent assay. The results from the acute stress model revealed that the individual extract of *Crataegus monogyna* decreased levels of TNF-α, while *Serratula coronata*, *Hypericum perforatum*, and *Valeriana officinalis* effectively reduced IL-6 levels. Both combinations, Antistress I and Antistress II, were effective in reducing TNF-α and IL-6 levels, with Antistress II also increasing IL-10 levels. In the chronic stress model, *Hypericum perforatum* extract decreased the levels of the pro-inflammatory cytokines TNF-α and IL-6, whereas extracts of *Serratula coronata* and *Valeriana officinalis* only reduced TNF-α levels. The two combined extracts, Antistress I and Antistress II, decreased TNF-α and IL-6 levels, while Antistress I also reduced the levels of the anti-inflammatory cytokine IL-10. The combinations of plant extracts used in our experiment have not been previously studied or documented in the available literature. However, based on our own experimental results, we can draw the conclusion that the combinations exhibit a more pronounced effect in reducing cytokine levels compared to the individual plant extracts.

## 1. Introduction

The current dinamic lifestyle is invariably associated with the development of stress, which negatively affects the normal functioning of many systems in the body and can trigger neuropsychiatric disorders, among which the most common are anxiety and depression. The term “stress”, defined as a non-specific reaction of the body to the requirements of the changed external environment, was first introduced by Hans Selye in 1936 [1]. Experimental and clinical studies have shown an association between oxidative stress, anxiety and depression, and changes in cytokine levels [2]. The pro-inflammatory cytokine Tumor necrosis factor-α (TNF-α) plays an important role in cell proliferation, differentiation, apoptosis and necrosis, and also influences stress-induced inflammatory changes [3]. Interleukin 6 (IL-6) is a cytokine that is involved not only in immune responses to inflammation and infection, but also regulates metabolic, regenerative, and neuronal processes [4]. As a result of exposure to cold stress, immobilization and forced swimming, an increase in IL-6 and TNF-α levels was observed in rats, with peak values reported 30 min after stress exposure [5]. This is one of the main models for causing depressive symptoms in rodents [6]. A correlation was found between elevated IL-6 levels in mice, increased immobility in the forced swimming test, and decreased sucrose consumption as positive indicators of depression [7]. A meta-analysis of various clinical trials in patients with depressive disorder shows a significant increase in the levels of IL-2, IL-6, TNF-α, IFN-γ compared to those in the control group. An increase in the concentration of proinflammatory cytokines is likely to be important for the development of depressive symptoms [8].

Interleukin 10 (IL-10) is an anti-inflammatory cytokine that is produced by Th2 cells and stimulates the proliferation and differentiation of autoreactive B cell lines. At the same time, it reduces Th1 activation and production of INF-γ. IL-10 antagonists have a good effect in the treatment of autoimmune diseases such as systemic lupus erythematosus. Stress has been shown to aggravate the course of the disease [9]. A correlation between distress, elevated IL-10 levels, and the development of malignancies has been suggested. IL-10 is elevated in forced swimming as an acute model of stress and immobilization as a chronic model of stress. Increased production of IL-10 has been observed in patients with melanoma and several types of lymphoma, and it is thought that it may affect the development of this type of malignancy [10]. Melanoma cells themselves cause overexpression of IL-10. It is also seen in squamous cell carcinomas such as bronchial carcinomas. T-cytotoxic cells, that are found in this type of tumor, proliferate only when IL-10 is neutralized by monoclonal antibodies. Increased production of IL-10 in gastrointestinal, lung, and other solid tumors has been associated with increased resistance to antineoplastic therapy and low patient survival [11].

Natural products, such as individual and combined plant extracts, contain a wide variety of biologically active substances (especially flavonoids) and are used to treat neuropsychiatric disorders such as anxiety and depression. Natural products are characterized by lower toxicity compared to synthetic antidepressants and anxiolytics [12].

The aim of the present study is to determine the effects of two combined plant extracts on the serum levels of the cytokines TNF-α, IL-6 and IL-10 in rats subjected to models of acute cold stress and chronic unpredictable stress. The effects of the two combined plant extracts are also compared with those of the individual plant extracts, which are part of the combined extracts. Some of the most common medicinal plants with proven antioxidant, anxiolytic and antidepressant action [13] have been used: *Valeriana officinalis*, *Melissa officinalis*, *Crataegus monogyna*, *Hypericum perforatum* and *Serratula coronata* in unique combinations which have not been studied yet. The ratios have been selected based on a recipe of Peter Dimkov—a well-known practitioner of herbal medicine in Bulgaria in the last century [14]. The authors’ team has developed two HPLC methods for determining the main active compounds in the extracts [15], has proved their non-toxicity and antioxidant activity [16], and has also investigated their antidepressant action [17].

## 2. Results

### 2.1. Chromatographic Determination of Individual and Combined Plant Extracts Composition

The first step was to establish qualitative and quantitative characterization of the dry extracs. The chromatographic analysis showed the presence of flavonoids (rutin, vitexin, hyperoside, quercetin, apigenin) and terpenes (valerenic acid and bornyl acetate), as well as 20-hydroxyecdyzone, hypericin and phenolic acids—rosmarinic, caffeic, p-coumaric, ferulic—in used extracts (Figure 1). The determined amounts of these components of the extracts are shown in Table 1.

In our experiment we used 72 Wistar rats divided in 9 groups (Table 2). After the treatment with extracts and the application of chronic stressors for 8 weeks, the animals were decapitated and their blood was collected. We measured the serum cytokines and compared their concentrations at the experimental groups with these of the control groups.

### 2.2. Changes in Serum TNF-α in Rats Subjected to Acute and Chronic Stress

According to our results, presented on Figure 2, a statistically significantly lower TNF-α levels were observed in the group of rats not subjected to stress compared to the control group exposed to the acute cold stress model. The groups treated with the combinations Antistress I and Antistress II exhibited a significant decrease in TNF-α levels when compared to the control group exposed to stress. There was a tendency towards a decrease in TNF-α levels in the groups treated with the individual extracts although the decrease was less pronounced. Only the *Crataegus monogyna* group showed statistical significance in reducing TNF-α levels.

Following chronic stress, TNF-α remained low in stress-free control animals and decreased significantly in animals of both groups treated with the Antistress I and Antistress II combinations compared to the control stress group with statistical significance (*p* < 0.001). In the groups treated with extracts of *Serratula coronata*, *Hypericum perforatum* and *Valeriana officinalis*, serum TNF-α also decreased statistically significantly compared to the results of the animals in the control stress group. The results of changes in serum TNF-α levels in chronically stressed rats are shown in Figure 3.

### 2.3. Changes in Serum IL-6 in Rats Subjected to Acute and Chronic Stress

In the stress-free control group, a statistically significant lower IL-6 level was observed compared to the control group of animals subjected to the acute cold stress model. Both combinations Antistress I and Antistress II showed a significant decrease in IL-6 levels compared to the control group of animals subjected to the acute cold stress model, with more pronounced changes observed in the Antistress I group. When examining the individual extracts, a decrease in IL-6 levels was observed in all of them compared to the control group of animals subjected to the acute cold stress model, and in the *Serratula coronata*, *Hypericum perforatum* and *Valeriana officinalis* groups, this decrease was statistically significant. The most significant decrease in IL-6 levels for the individual extracts was observed in the *Serratula coronata* group. These results are shown in Figure 4.

When analyzing the effects of the chronic stress model, we observed that serum IL-6 level remained low in stress-free control animals. In the groups of animals treated with the combined Antistress I and Antistress II extracts, IL-6 levels decreased significantly compared to the stress control group. Among the individual plant extracts, only the group treated with *Hypericum perforatum* showed a statistically significant decrease in IL-6 levels compared to the stress control group. Figure 5 illustrates the results of the changes in serum IL-6 levels in rats exposed to chronic stress.

### 2.4. Changes in Serum IL-10 in Rats Subjected to Acute and Chronic Stress

As seen in Figure 6, IL-10 levels in the serum were lower in the non-stress control group compared to the control stress group subject to the acute stress, although this difference was not statistically significant. Among the groups treated with the combinations, IL-10 levels decreased slightly in the Antistress I group and increased in the Antistress II group. The increase in the level of IL-10 in the Antistress II group compared to the stress control group was statistically significant. When examining the individual extracts, only the *Serratula coronata* group exhibited a mild decrease in IL-10 levels, which was not statistically significant. In all other groups, IL-10 levels increased, and in the *Melissa officinalis* group, this increase was statistically significant.

The results of changes in serum IL-10 levels in chronically stressed rats are shown in Figure 7. In our study, serum IL-10 levels are significantly lower in the stress-free control group versus the control group subject to chronic stress. Of the two combinations, only Antistress I decreased the studied indicator by statistical significance, while in the groups treated with the Antistress II combination and with individual extracts of *Crataegus monogyna*, *Hypericum perforatum* and *Seratula coronata*, the changes were not significant and only a tendency to decrease the indicator was observed. In the groups treated with *Melissa officinalis* and *Valeriana offcinalis* there is a tendency to increase IL-10 compared to the control stress group.

## 3. Discussion

### 3.1. Changes in Serum TNF-α in Rats Subjected to Acute and Chronic Stress

The results from our study showed a statistically significant (*p* < 0.05) increase in the serum levels of the pro-inflammatory cytokine TNF-α in rats subjected to an acute cold stress model. Similar outcomes have been reported in several other studies [5]. Himmerich et al. have used Forced swim test as an acute stress model and observed an elevated production of TNF-α [10]. Another study, similar to our experimental setup with acute cold stress, showed increased levels of TNF-α and IL-6 in the blood serum of Wistar rats [18]. Notably, there exists a connection between anxiety and increased levels of TNF-α following acute stress [19].

High levels of TNF-α as a result of chronic stress were also observed in a study by Li et al. [20]. TNF-α gene expression is stimulated by subchronic immobilization stress and this correlates with decreased cerebral neutrophic factor in the hippocampus [21]. Binding to the TNF receptor 1 activates enzymes that stimulate the release of free radicals involved in the pathogenesis of depression [22]. The dry extract of *Serratula coronata* at a dose of 500 mg/kg bw veraciously reduces the serum levels of TNF-α in the animals we studied. The main biologically active component in the plant is the ecdysteroid 20-hydroxyecdisone, whose concentration is 18,928 μg/g dw Table 1. It is known to enhance cognitive functions by promoting antioxidant actions within the brain [23]. Yang and co-authors found that a dry extract of *Achyranthes japonica* roots containing 20-hydroxyecdisone inhibited the expression of genes encoding TNF-α synthesis [24]. This may be due to the antioxidant action of the substance and inhibition of NF-κB activation, which in turn inhibits TNF-α production [25]. 20-Hydroxyecdisone has a protective effect against lipopolysaccharide-induced lung damage in mice by inhibiting the expression of genes for the synthesis of proinflammatory cytokines TNF-α, IL-2, IL-6 and IL-8 and induces the expression of genes for the synthesis of anti-inflammatory cytokines IL-4 and IL-10 [26]. Treatment with 20-hydroxyecdisone reduced TNF-α, IL-1β, IL-6 and NF-κB in rat serum in a rheumatoid arthritis model [27]. The anxiolytic effect of *Valeriana officinalis* is attributed to valepotriates and valerenic acid, which cause an increase in Gamma-aminobutyric acid (GABA) within the central nervous system [28]. It has also procognitive effect is linked to its antioxidant activity in the hippocampus and the enhancement of GABA neurotransmission, which stimulates the dendrite outgrowth of newly formed neurons [29]. Regarding the effect of *Valeriana officinalis* on TNF-α levels, valerenic acid, one of the main biologically active components of *Valeriana officinalis*, has been found to reduce neuroinflammation and levels of proinflammatory cytokines, including TNF-α and IL-6 in brain homogenate in a model of Parkinson’s disease [30]. Another study showed that the use of Imipramine and diene valepotriates derived from *Valeriana glechomifolia* reduced depressive-like behavior in animals and decreased TNF-α levels [31]. Similar to our study, Hatano et al. studied the effect of *Hypericum perforatum’s* extract at doses of 125, 250 and 500 mg/kg bw on depressive-like behavior and levels of TNF-α and corticosterone in the serum and hippocampus of Wistar female rats subjected to ovariectomy. The authors found a reduction in the immobility time in the forced swimming test at a dose of 500 mg/kg bw from the extract and dose-dependent decrease in TNF-α and corticosterone levels in both serum and hippocampus of animals [32]. The mechanisms by which the extract exerts its effects are by inhibition of NF-κB and its signaling pathway, leading to decreased expression of TNF-α [33]. In their study, Chen et al. found that hyperoside—the main biologically active ingredient in *Hypericum perforatum*—dose-dependently reduces the levels of TNF-α and IL-6 in brain homogenate in rats, in a model of diabetes mellitus through antioxidants, anti-inflammatory and anti-apoptotic mechanisms. The doses used orally were 50, 100 and 200 mg/kg bw [34], while in our study, the concentration of hyperoside contained in the Antistress II extract was 8161 μg/g dw, which is approximately 4 mg/kg bw administered orally to rats. *Hypericum perforatum’s* extract, like the antidepressants Milancipran and Agomelatin, decreased serum levels of cortisol and TNF-α, as well as depressive-like behavior in rats subjected to a subclinical hypothyroidism model [35].

### 3.2. Changes in Serum IL-6 in Rats Subjected to Acute and Chronic Stress

A study conducted by Himmerich et al. observed an increase in IL-6 levels in rats subjected to acute immobilization stress. Additionally, data from two other studies indicated an elevation in serum IL-6 levels in Wistar rats subjected to cold stress [10,18,19,20,21,22,23,24,25,26,27,28,29,30,31,32,33,34,35,36]. There is a correlation between increased IL-6 levels and the development of anxiety in experimental animals. A study by Jang et al. found higher levels of IL-6 and TNF-α in the serum of mice subjected to acute immobilization stress. The stressed animals also spent less time in the open arms of the elevated plus maze, indicating increased anxiety-like behavior [37]. Our study results demonstrate that both combinations of extracts significantly reduce the elevated levels of IL-6 induced by acute cold stress. Notably, both combinations include *Serratula coronata* extract. When administered individually, *Serratula coronata* also exhibits a significant reduction in serum IL-6 levels compared to the control group subjected to acute stress. This effect is likely attributed to the adaptogenic properties of *Serratula coronata*, as discussed in another study [38]. The adaptogens’ ability to enhance stress resistance diminishes the activation of the pituitary-hypothalamus-adrenal axis and the release of catecholamines, consequently lowering IL-6 levels. Both combinations also contain dry extract of *Valeriana officinalis*. In our study, administering the individual Valeriana extract leads to a significant decrease in IL-6 compared to the stressed control group. The effect of *Valeriana officinalis* on IL-6 production during acute stress is likely not due to its impact on catecholamines but rather its inhibition of NF-κB, which subsequently reduces the release of pro-inflammatory cytokines [39]. In our experiment, the individual administration of the dry extract of *Hypericum perforatum* (present in the Antistress II combination) significantly reduced IL-6 levels compared to the control group subjected to acute stress. Similarly, in another study, the oral administration of a water-ethanol extract of *Hypericum perforatum* at a dosage of 110 mg/kg b.w. in mice also resulted in decreased IL-6 levels [40]. The mechanism of action is not attributed to its effect on neurotransmitter levels but rather to the inhibition of genes encoding the release of COX-2 and inducible nitric oxide synthase [41]. The reduction of these enzymes is correlated with the suppression of NF-κB, which in turn leads to a decrease in pro-inflammatory cytokines [32,33,34,35,36,37,38,39,40,41,42].

In a preclinical study, the use of a chronic stress model caused by mild unpredictable stressors in mice resulted in depressive-like symptoms detected by behavioral tests and elevated IL-6 and Adrenocorticotropic hormone levels. The use of antidepressants normalizes the observed changes [43]. The decrease in IL-6 when using the combined extract Antistress I is probably due to the high content of phenolic acids, of which in the highest concentration—21,021 μg/g dw, is rosemarinic acid Table 1. When found alone in peripheral mononuclear cells, in amounts of 10, 20, 40 and 80 μM, it has been shown to reduce oxidative stress and cytokine levels such as TNF-α and IL-6 [44]. Regarding Antistress II, the observed decrease in IL-6 is probably due to the high content of flavonoids in the extract such as: rutin (21,582 μg/g dw), hyperoside (8161 μg/g dw) and quercetin (3366 μg/g dw). The extract also contains 20-hydroxyecdisone (1498 μg/g dw), hypericin (378 μg/g dw) and valerenic acid (395 μg/g dw) Table 1. Flavonoids can be used to treat neuropsychiatric disorders. For example, rutin has a neuroprotective and anti-inflammatory effect in an experimental model of rat brain damage [45]. The substance has a beneficial effect on a number of diseases of the nervous system, including depression [46]. Hyperoside also exhibits the antidepressant activity described in several experimental studies [47]. It also enhances spatial learning and memory in rats that are chronically stressed, as demonstrated by the Morris water maze test. This effect is linked to the substance’s antidepressant action and the stimulation of hippocampal expression of brain-derived neurotrophic factor (BDNF) [48]. Quercetin has anti-inflammatory and neuroprotective effects in several animal models of traumatic brain injury, anxiety and depressive disorders [49]. Grundmann and co-authors administered *Hypericum perforatum* extract at doses of 250 to 750 mg/kg bw to rats which were subjected to a chronic model of immobilization stress. Animals develop depressive-like behavior as detected by the open field test and increase serum IL-6 and TNF-α levels. Administration of *Hypericum perforatum* extract reduces elevated levels of both TNF-α at all doses and IL-6 at doses of 500 and 750 mg/kg bw [50]. A study in a model of experimental autoimmune encephalomyelitis in mice found that *Hypericum perforatum* extract in combination with gold nanoparticles decreased levels of IFN-γ, IL-17A and IL-6 and increased those of TGF-β, IL-10 and IL-4 [51]. In their study, Zhai et al. reported that hypericin, an active ingredient in *Hypericum perforatum*, reduced serum levels of IL-6, IL-1β and TNF-α in rat serum in a model of postpartum depression [52].

### 3.3. Changes in Serum IL-10 in Rats Subjected to Acute and Chronic Stress

Acute stress induced by inescapable electric shocks increased IL-10 and TNF-α levels in mouse plasma [53]. Conversely, Kalinichenko et al. discovered that acute immobilization stress decreased plasma IL-10 levels in rats displaying active behavior, in contrast to those with passive behavior. Changes in plasma cytokine levels were more pronounced in active animals [54]. Acute stress induced by surgical manipulation increased TNF-α levels but had no effect on IL-10 levels [55]. In our study, Antistress II significantly increased IL-10 levels compared to the control group of animals subjected to acute stress. However, the individual extracts of *Hypericum perforatum* and *Valeriana officinalis* included in the combination exhibited only a tendency to increase IL-10 without statistical significance (*p* > 0.05). In another study, *Hypericum perforatum* extract increased serum IL-10 levels in mice when administered for five days, potentially due to its anti-inflammatory properties [56]. The individual extract of *Melissa officinalis* increased IL-10 levels in the serum of rats subjected to acute cold stress compared to the control group. We did not find any other studies on the effect of lemon balm extract on IL-10 levels in the available data. The primary biologically active compound in the plant, rosmarinic acid, increased IL-10 only in an in vitro model of oxidative stress [57].

As for chronic stress models, The Freitas study showed that physical stress due to strenuous physical training in rats for 6 weeks showed animal adaptation and decreased levels of a number of cytokines, including IL-10 [58]. In a study by Mesquita et al., increasing IL-10 levels in female mice reduced depressive behavior, while no changes in IL-10 levels were found in male mice [59]. According to another study, chronic stress increases IL-10 levels [60], which is consistent with our results. Manikowska and co-authors studied the change in TNF-α, IL-6 and IL-10 in Wistar rats subjected to chronic stress caused by mild stressors for 6 weeks. Increases in serum cytokine levels have been observed with the use of the antidepressant Mianserin to normalize elevated levels of all 3 cytokines [61]. The direction and extent of changes in cytokine levels depend on the type and duration of the stressor, the post-stress period and other exogenous factors [62]. One-time stress has an immunostimulatory effect, while prolonged stress suppresses the immune system and is associated with the body’s inability to adapt to changing environmental requirements [63]. This may cause a permanent pathological increase in IL-10 levels.

### 3.4. Link between Oxidative Stress, Psychological Disorders and Elevated Levels of Proinflammatory Cytokines

Oxidative stress is an important factor associated with the development of anxiety and depression and elevated levels of proinflammatory cytokines. The use of antioxidants has been shown to lower levels of TNF-α, IL-6 and other proinflammatory cytokines [44]. The relationship between oxidative stress and inflammation is two-way one. On one hand, proinflammatory cytokines stimulate the release of free radicals to counteract pathogens, on the other hand, oxidative stress leads to inflammation and increases the release of proinflammatory cytokines [64]. The etiology of depression is not well understood, but oxidative stress is considered to be one of the main factors contributing to its development, along with genetic predisposition and psycho-emotional stress [65]. Cell-mediated immune activation and inflammation contribute to the development of depressive symptoms such as anhedonia, anxiety-like behavior, fatigue, and others. These effects are partly mediated by elevated levels of proinflammatory cytokines such as TNF-α and IL-6 and oxidative stress. Activation of free radicals as well as suppression of antioxidant systems have been observed [66]. Therefore, the reduction in proinflammatory cytokines from the combined extracts in the present study is related to the results of our previous studies on their antioxidant activity [16], as well as influencing depressive-like behavior in rats in a chronic stress model [17].

### 3.5. Benefits of Using Combinations of Plant Extracts

The presence of many biologically active ingredients in medicinal plants is a prerequisite for various desired and adverse interactions. The induction of cytochrome P450 subtypes, which is caused by *Hypericum perforatum*, affects the metabolism of some drugs such as anticoagulants, chemotherapeutics, statins, conventional antidepressants, etc., which can lead to life-threatening effects [12]. On the other hand, with an appropriate combination of plant extracts, potentiation and synergy of their pharmacological action can be observed. In our study, the individual extract of *Serratula coronata* led to a reduction in the cytokines TNF-α and IL-6. When used in the Antistress I and Antistress II combinations, the required dose to achieve this effect was 10 times less. This gives us reason to assume a synergistic effect of *Serratula coronata* extract in the combination. When using combined extracts, the active substances are in lower therapeutic doses than when using the single plant extracts. Therapeutic indications are expanding, and new effects can be observed that have not been described at the use of herbal extracts alone [67]. A study by Muller and co-authors compared the therapeutic effect of the combination of dry extracts of *Hypericum perforatum* and *Valeriana officinalis* with that of dry extract of *Hypericum perforatum* used alone in the treatment of depression combined with anxiety. The presence of *Valeriana officinalis* in the combined extract leads to a rapid response to the anxiety component, expressed by tension, dysphoria, and sleep disorders. The treatment of these symptoms is important because they play a major role in lowering the quality of life of patients suffering from depressive disorders [68]. The combination of extracts of *Crataegus oxyacantha*, *Valeriana officinalis* and *Passiflora incarnata* showed a pronounced anxiolytic effect on experimental animals, as evidenced by the raised cross maze test, and also improved the learning and memory processes reported in the step-through apparatus [69]. The combination of aqueous-ethanolic extracts of *Valeriana officinalis*, *Melissa officinalis* and *Passiflora incarnata* was studied for anxiolytic activity in the elevated plus maze test and social interaction in rats. A synergistic interaction is observed in the combination, as the reduction in anxiety is greater and is achieved at lower doses compared to plant extracts alone [70].

## 4. Materials and Methods

### 4.1. Plant Extracts

Dry extracts of Valerian (*Valeriana officinalis*, *Valerianaceae*) root, Lemon balm (Melissa officinalis, Lamiaceae) herb, Hawthorn (*Crataegus monogyna*, *Rosaceae*) flower and leaf, St. John’s wort (*Hypericum perforatum*, *Hypericaceae*) herb and Seratula (*Serratula coronate*, *Asteraceae*) herb, and combined extracts based on them and known as Antistress I and Antistress II were used. Antistress I is a combination of *Valeriana officinalis*, *Melissa officinalis*, *Crataegus monogyna* and *Seratula coronata* in proportion of 4:3:3:1, and Antistress II—Valeriana officinalis, *Hypericum perforatum* and *Seratula coronata* in proportion of 4.5:4.5:1. The dry extracts are provided by AVICENA HERB Ltd., Smolyan, Bulgaria. They were obtained by extraction of the drug with 40% ethanol and subsequent spray drying according to the industrial technology of EXTRACTPHARMA Ltd., Sofia, Bulgaria by order of AVICENNA HERB Ltd., Smolyan, Bulgaria. The compositions of the used extracts were determined by HPLC methods developed by Katsarova et al. [15]. The HPLC system was composed of a ProStar 230 solvent delivery module and photo diode array detector model 335, for Method I: Hitachi C18 AQ column (250 mm × 4.6 mm, 5 μm). A solvent system including H_2_O (A) with pH 3.7 (adjusted with H_3_PO_4_) and acetonitrile (B) was used in gradient condition from 90A:10B to 10A:90B. The flow rate was 0.9 mL/min and detection at 335 nm for fenolic acids and flavonoids. For Method II: Microsorb-MV C18 column (150 × 4.6 mm, 5μm), a solvent system—H_2_O (A) with pH 3.7 and acetonitrile:methanol 1:1 (B) in gradient condition from 80A:20B to 10A:90B, flow rate—1 mL/min and detection at 245 nm for 20-hydroxyecdyzone, 210 nm for bornyl acetate, valerenic acid and 285 nm for hypericin. The compounds of interest in the extracts were identified through their retention times as well as by comparing their absorbtion spectra with those of standard substances.

### 4.2. Experimental Animals and Treatment

Male Wistar rats with an average weight of 180–200 g were used. The animals were kept under standard laboratory conditions: 12:12 h dark/light cycle, 45% relative humidity, temperature 26.5 °C ± 1 °C, and free access to food and water. The experiments were approved by the Commission on Animal Ethics of the Bulgarian Food Safety Agency with permission № 127 from 9 December 2015 and decision of the Ethics Commission at MU—Plovdiv with protocol № 3 from 21 April 2016. We used a total of 144 animals. For each experiment we used 72 rats, divided into 9 groups of 8 animals (Table 2).

The animals of all groups were treated daily orally by gavage with both combinations and the individual extracts included in their composition. Prior to oral administration, the dry extracts were dissolved in distilled water. The duration of the treatment was 15 days for the acute stress experiment and 8 weeks for the chronic stress experiment. 1 h after the last stressor, blood was collected from the animals by decapitation under etheric anesthesia in a glass bell filled with diethyl ether vapor for 60 s.

### 4.3. Acute Cold Stress Model

In this experimental model, the stressor is applied once on the 15th day after the start of the experiment. Following the administration of the extracts, all rats, except those in the stress-free control group, underwent acute cold stress. This involved placing them in plastic cages and exposing them to a refrigerated environment set at 4 °C for a duration of 12 h. The purpose of implementing the acute cold stress model in this study is to induce anxiety-like responses in the experimental animals. This method of acute stress is according to the study of Guo et al. [18].

### 4.4. Chronic Stress Model

In this model, soft stressors are applied daily, which are: inclination of the cell at 45° for 24 h, wetting of chips 200 mL of water per 100 g of chips for 24 h, flashing light (60 flashes per minute) for 3 h, sounds of natural predator (recording of cat sounds) for 15 min, deprivation of water for 24 h followed by limited access to food, deprivation of water for 24 h, followed by placing an empty bottle for 1 h. To avoid adaptation to stressors, each of them is applied on a different day each week. The model of chronic stress that we have chosen is suitable for experimental studies of anti-stress effects and is well valid because it mimics the unpredictability of everyday stressors. It is one of the most used models for inducing chronic stress which also affect systemic inflammation leading to alterations in serum cytokine levels [71].

### 4.5. In Vitro Study of Pro- and Anti-Inflammatory Cytokines

The proinflammatory cytokines TNF-α and IL-6 and the anti-inflammatory cytokine IL-10 in rat serum were tested by commercially available ELISA kits (Platinum ELISA, eBioscience, Waltham, MA, USA) in strict compliance with the manufacturer’s guidelines.

The method used is the Enzyme-linked immunosorbent assay (ELISA). Principle of the method: for quantitative cytokine testing, diluted rat serums (1:2) for IL-6, IL-10 and TNF-α, internal controls and test standards were dripped on solid phase with monoclonal antibodies against the corresponding cytokine. After incubation and washing, a peroxidase conjugate (second anti-species antibody) is placed to form a cytokine complex. A second wash follows to remove unbound conjugate. When a chromogenic substrate is added to the enzyme, a color reaction occurs, marking the presence of cytokines. Absorption, which is proportional to the cytokine concentration, is measured colorimetrically on a TECAN ELISA reader at 450 and 620 nm. The concentration of each cytokine in pg/mL is determined by plotting a standard curve.

### 4.6. Statistical Analysis

The statistical processing of the obtained results is performed with the software product IBM SPSS 19.0. A Shapiro-Wilk test is performed to determine the distribution. An arithmetic mean (mean) and a standard error of the arithmetic mean (±SEM) are determined for each indicator. Comparison of results between groups is performed with One way ANOVA, followed by LSD (least significant difference) post hoc test to identify significant differences between the groups. Results at a significance level of *p* < 0.05 are considered statistically significant.

## 5. Conclusions

Our study findings indicate that in an acute cold stress model, both combinations Antistress I and Antistress II effectively reduce the levels of proinflammatory cytokines TNF-α and IL-6, with Antistress II additionally increasing IL-10 levels. The individual extracts predominantly reduce IL-6 levels, except for *Crataegus monogyna*, which specifically reduces TNF-α levels. In chronic stress model, the combined extracts Antistress I and Antistress II effectively reduced the levels of both TNF-α and IL-6, and Antistress I also demonstrated a reduction in the levels of the anti-inflammatory cytokine IL-10. Among the individual extracts, the *Hypericum perforatum* extract successfully decreased the levels of proinflammatory cytokines TNF-α and IL-6, while *Serratula coronata* and *Valeriana officinalis* extracts only reduced TNF-α levels. These effects can be attributed to the proven antioxidant activity of the combined extracts and their ability to influence anxiety and depressive-like behavior in rats subjected to acute cold stress and chronic unpredictable stress. Based on these findings, we can suggest that after clinical trials, Antistress I and Antistress II, which are new combinations that contain carefully selected medicinal plants in the proposed ratios, may be utilized for the prevention and treatment of anxiety and depression.

## Figures and Tables

**Figure 1 plants-12-03049-f001:**
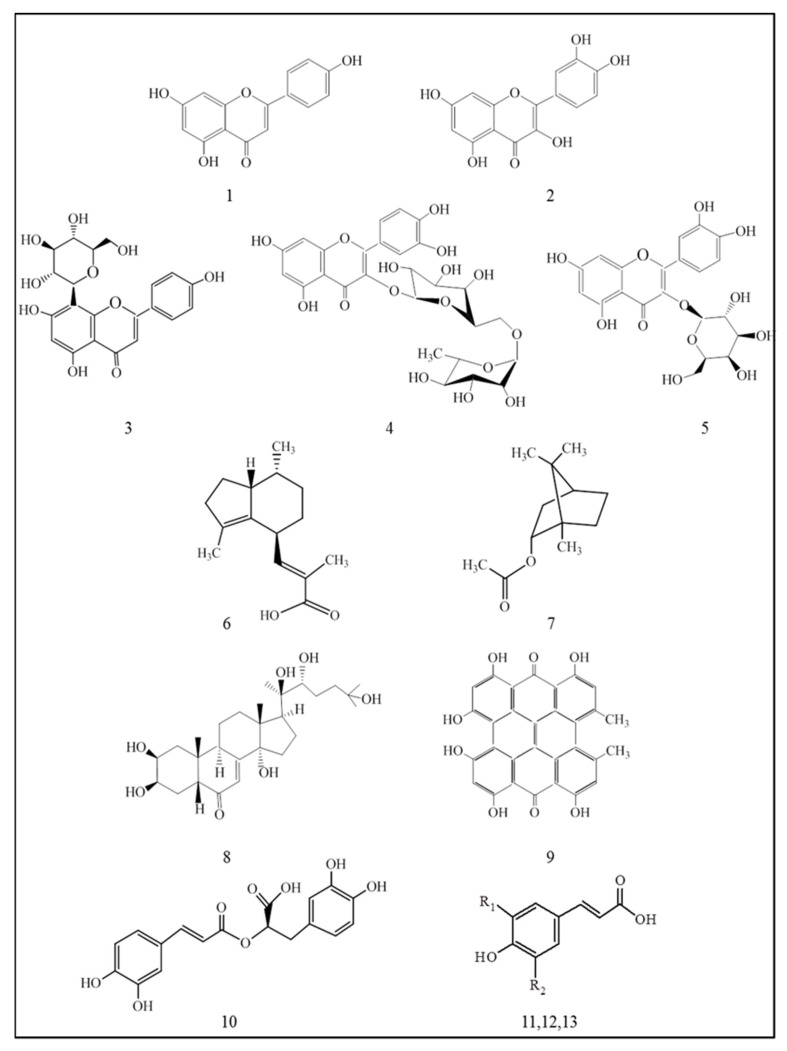
Biologically active substances in the studied extracts—flavonoid aglycones (apigenin-1, quercetin-2); flavonoid glycosides (vitexin-3, rutin-4, hyperoside-5); terpenes (valerenic acid-6, bornyl acetate-7); phytosteroid—20-hydroxyecdyzone-8; anthraquinone-derivative—hypericin-9; phenolic acids (rosmarinic acid-10, p-coumaric acid-11 R_1_=H R_2_=H, caffeic acid-12 R_1_=OH R_2_=H, ferulic acid-13 R_1_=OCH_3_ R_2_=H).

**Figure 2 plants-12-03049-f002:**
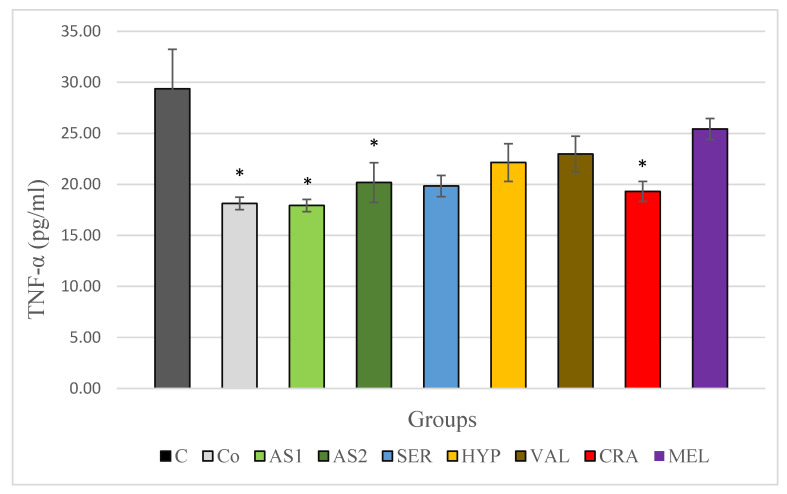
Serum TNF-α in Wistar rats (8 in each group) subjected to acute stress and treated for 14 days with 2 combinations and 5 individual plant extracts. Comparison was made by using ANOVA test, followed by LSD post-hoc test * *p* < 0.05 compared to stress control.

**Figure 3 plants-12-03049-f003:**
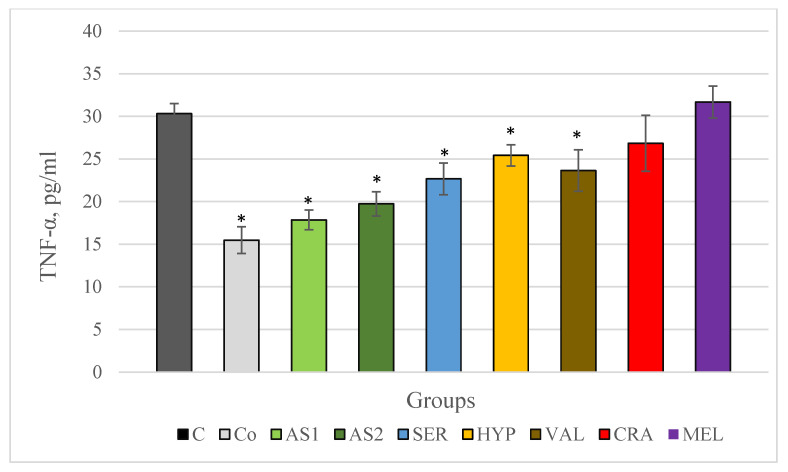
Serum TNF-α in Wistar rats (8 in each group) subjected to chronic stress and treated for 8 weeks with 2 combinations and 5 individual plant extracts. Comparison was made by using ANOVA test, followed by LSD post-hoc test * *p* < 0.05 compared to stress control.

**Figure 4 plants-12-03049-f004:**
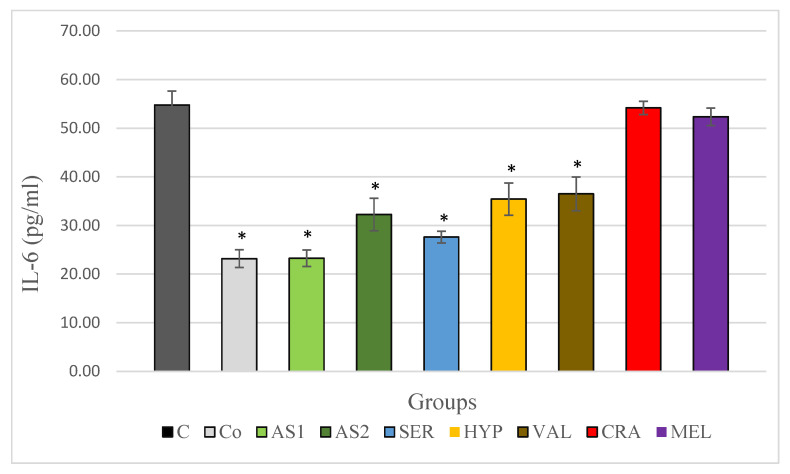
Serum IL-6 in Wistar rats (8 in each group) subjected to acute stress and treated for 14 days with 2 combinations and 5 individual plant extracts. Comparison was made by using ANOVA test, followed by LSD post-hoc test * *p* < 0.05 compared to stress control.

**Figure 5 plants-12-03049-f005:**
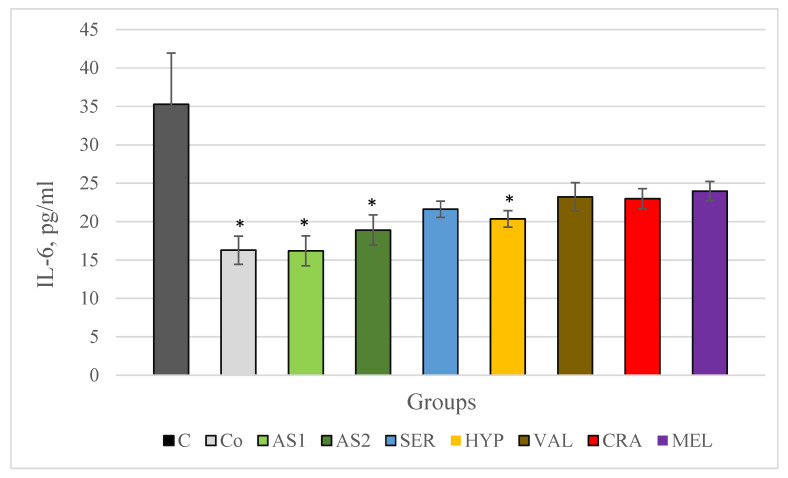
Serum IL-6 in Wistar rats (8 in each group) subjected to chronic stress and treated for 8 weeks with 2 combinations and 5 individual plant extracts. Comparison was made by using ANOVA test, followed by LSD post-hoc test * *p* < 0.05 compared to stress control.

**Figure 6 plants-12-03049-f006:**
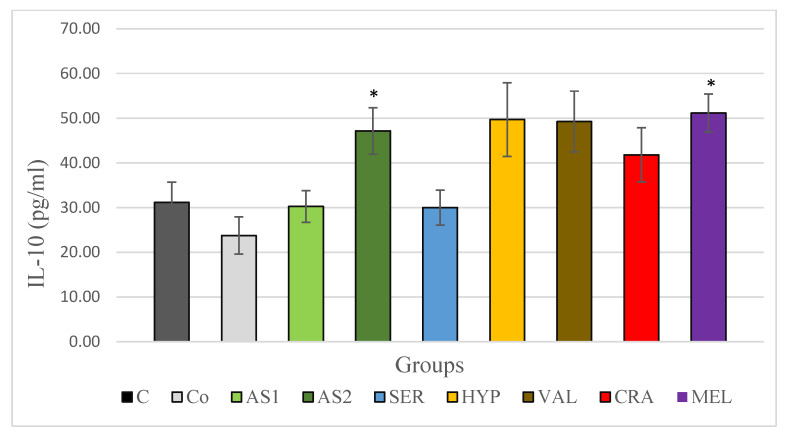
Serum IL-10 in Wistar rats (8 in each group) subjected to acute stress and treated for 14 days with 2 combinations and 5 individual plant extracts. Comparison was made by using ANOVA test, followed by LSD post-hoc test * *p* < 0.05 compared to stress control.

**Figure 7 plants-12-03049-f007:**
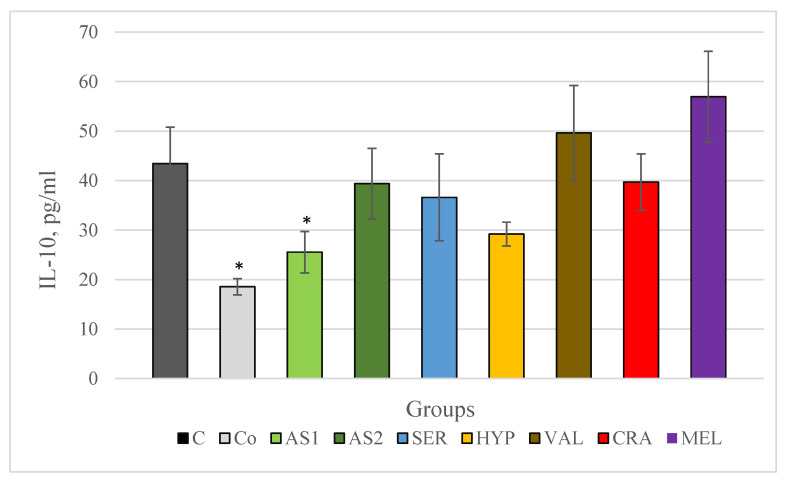
Serum IL-10 in Wistar rats (8 in each group) subjected to chronic stress and treated for 8 weeks with 2 combinations and 5 individual plant extracts. Comparison was made by using ANOVA test, followed by LSD post-hoc test * *p* < 0.05 compared to stress control.

**Table 1 plants-12-03049-t001:** Quantities of bioactive constituents identified in μg/g dry extract *.

	Extracts	*V. officinalis*	*M. officinalis*	*C. monogyna*	*S. coronata*	*H. perforatum*	Antistress I	Antistress II
Analyte, µg/g								
Caffeic acid		53 ± 5	985 ± 28	trace	nd	nd	392 ± 20	nd
Rutin		45 ± 4	232 ± 6	1623 ± 106	nd	40,832 ± 299	549 ± 7	21,582 ± 119
Vitexin		nd	245 ± 8	764 ± 31	trace	nd	323 ± 7	nd
Hyperoside		nd	524 ± 19	893 ± 39	615 ± 29	15,538 ± 251	385 ± 6	8161 ± 199
p-Coumaric acid		70 ± 4	48 ± 3	trace	nd	trace	32 ± 1	nd
Ferulic acid		35 ± 2	198 ± 5	trace	nd	trace	85 ± 1	nd
Quercetin		nd	trace	6100 ± 212	5320 ± 105	5985 ± 111	2250 ± 71	3366 ± 301
Apigenin		nd	trace	trace	814 ± 29	1665 ± 29	81 ± 2	366 ± 8
Rosmarinic acid		1020 ± 41	58,320 ± 334	3250 ± 81	nd	nd	21,021 ± 93	trace
20-hydroxyecdyzone		nd	nd	nd	18,928 ± 295	nd	1235 ± 79	1498 ± 88
Hypericin		nd	nd	nd	nd	905 ± 41	nd	378 ± 18
Bornyl acetate		69 ± 3	nd	nd	nd	nd	trace	trace
Valerenic acid		825 ± 24	nd	nd	nd	nd	258 ± 11	395 ± 19

* Values are means ± standard deviation of triplicate samples; nd, not detected.

**Table 2 plants-12-03049-t002:** Distribution of experimental animals by groups.

Group	Legend	Description	Stress
1	C	Distilled water 10 mL/kg body weight	Yes
2	C_0_	Distilled water 10 mL/kg body weight	No
3	AS1	Antistress I 500 mg/kg body weight	Yes
4	AS2	Antistress II 500 mg/kg body weight	Yes
5	SER	*Serratula coronata* 500 mg/kg body weight	Yes
6	HYP	*Hypericum perforatum* 500 mg/kg body weight	Yes
7	VAL	*Valeriana officinalis* 500 mg/kg body weight	Yes
8	CRA	*Crataegus monogyna* 500 mg/kg body weight	Yes
9	MEL	*Melissa officinalis* 500 mg/kg body weight	Yes

## Data Availability

The data presented in this study are available on request from the corresponding author.

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
