# Peer review of "Effect of Plant Extracts Combinations on TNF-α, IL-6 and IL-10 Levels in Serum of Rats Exposed to Acute and Chronic Stress"

_plants, 2023, doi:10.3390/plants12173049_

Round 1

Reviewer 1 Report

The purpose of this study is to look at how two flavonoid-rich plant extracts combined affect the levels of the cytokines TNF-α, IL-6, and IL-10 in rats that have been exposed to models of acute cold stress and long-term unpredictable stress. Common medicinal plants like Valeriana officinalis, Melissa officinalis, Crataegus monogyna, Hypericum perforatum, and Serratula coronata were used in the study, and they were mixed into two distinct combinations called Antistress I and Antistress II. TNF-α and IL 6 levels were reduced by the two combined extracts, Antistress I and Antistress II, and Antistress I also decreased levels of the anti-inflammatory cytokine IL 10.

This study is interesting and important, but there are some important notes

 before the manuscript can be considered for publication.

My comments are as the following:

1-First, the authors need to confirm the relationship of inflammation to stress in this experiment by measuring stress hormones such as corticosteroids and their relationship to cytokines.

2-Despite the use of plants, the authors did not measure the effect of plants on oxidative stress or the antioxidant effect in order to be able to interpret the results of the research.

Specific comments:

Introduction

1-There is a need for an introduction to the importance of the plants selected for the experiment, Melissa officinalis, Crataegus monogyna, Hypericum perforatum and Serratula coronata, what is their traditional use and their relationship to depression and inflammation, and what is the family of each plant?

Methods

1-Where was the plant collected from, who identified it, and what is the identification number of the herbarium.

2- Mention in detail how the active substances present in each plant were identified, what equipment and method was used for separation and assessment

3-What is the method used to extract the plants and the solvents used and their percentage.

4- On what basis were plant doses chosen for this experiment, where was the toxicity test?

5- How was the extract dissolved and diluted to be given to animals?

6-Statistical analysis is incomplete What is the analysis that was used after the ANOVA to find out the significant differences between the groups and the differences between all groups must be calculated

Results

1-All Figures must contain the number of rats that were used for each group with the determination of the statistical analysis ANOVA and after ANOVA and the presence of signs on the columns showing the significant difference between all groups.

 The paper results well written and comprehensive. Just some little typing errors.

Author Response

The authors thank the reviewers for their comments and suggestions, which aim to improve the quality of the manuscript.

Reply to reviewer  1:

1-First, the authors need to confirm the relationship of inflammation to stress in this experiment by measuring stress hormones such as corticosteroids and their relationship to cytokines.

Indeed, the relationship between inflammation and stress, as well as the associated changes in cytokine levels, is well-established in scientific literature. Several sources that we have cited in our manuscript (e.g., refs. 3, 8, 12) provide substantial evidence supporting this relationship. Furthermore, the relationship between corticosteroids and cytokines is a thoroughly studied area. Since this relationship is well-established, we deemed it unnecessary to include it in our current study. Our research focus was rather on the specific effects of plant extracts on cytokine levels, in alignment with our broader research goals.

We appreciate your insight, and we hope this clarifies the scope and orientation of our study.

2-Despite the use of plants, the authors did not measure the effect of plants on oxidative stress or the antioxidant effect in order to be able to interpret the results of the research.

We would like to clarify that the present work is part of a series of studies our research team has conducted on the quality and effects of the extracts in question. Specific HPLC methods have been developed to standardize the extracts of each medicinal plant based on its characteristic active substances, as referenced in our manuscript (ref. 15). Moreover, prior studies have demonstrated their antioxidant activity and lack of toxicity (ref. 16), as well as their antidepressant effect (ref. 17).The focus of the current study is on examining the effects of the combinations on cytokine levels. Based on the results obtained, we propose the Antistress 1 and 2 combinations for use in the prevention and treatment of anxiety and depression, pending further clinical trials.

Specific comments:

Introduction

1-There is a need for an introduction to the importance of the plants selected for the experiment, Melissa officinalis, Crataegus monogyna, Hypericum perforatum and Serratula coronata, what is their traditional use and their relationship to depression and inflammation, and what is the family of each plant?

The medicinal plants mentioned are very well known and widely used, and we have mentioned their effects very briefly (line 87). We add ref. 13. - A. Sharma, A. Cardoso-Taketa, G. García, M. Villarreal, A systematic updated review of scientifically tested selected plants used for anxiety disorders Botanics: Targets and Therapy, 2, 21 (2012).

So also the families of each of the plants have been added (part 4.1) according to your recommendation. The use of medicinal plants in depression is mentioned in lines 78-81 (ref. 12) and in more detail in part 3.5.

Methods

1-Where was the plant collected from, who identified it, and what is the identification number of the herbarium.

We have used individual and combined dry extracts provided by the company Avicena Herb EOOD, Smolyan and made by the company Ekstraktfarma EOOD, Sofia, which guarantees the quality of the raw material. In this regard, we add in part 4.1. the following sentence: They were obtained by extraction of the drug with 40% ethanol and subsequent spray drying using the industrial technology of Extractpharma, Ltd., Sofia, Bulgaria for Avicenna Herb Ltd., Smolyan, Bulgaria.

2- Mention in detail how the active substances present in each plant were identified, what equipment and method was used for separation and assessment

The following text was added in part 4.1:

The HPLC system was composed of a ProStar 230 solvent delivery module and photo diode array detector model 335, for Method I: Hitachi C18 AQ column (250 mm × 4.6 mm, 5 μm). A solvent system including H2O (A) with pH 3.7 (adjusted with H3PO4) and acetonitrile (B) was used in gradient condition from 90A:10B to 10A:90B. The flow rate was 0.9 ml/min and detection at 335 nm for fenolic acids and flavonoids. For Method II: Microsorb-MV C18 column (150×4.6 mm, 5μm), a solvent system – H2O (A) with pH 3.7 and acetonitrile:methanol 1:1 (B) in gradient condition from 80A:20B to 10A:90B, flow rate – 1 ml/min and detection at 245 nm for 20-hydroxyecdyzone, 210 nm for bornyl acetate, valerenic acid and 285 nm for hypericin. The compounds of interest in extracts were identified through their retention times as well as by comparing their absorbtion spectra with those of standard substances.

3-What is the method used to extract the plants and the solvents used and their percentage.

The extraction was carried out with 40% ethanol. The information has been added to the text.

4- On what basis were plant doses chosen for this experiment, where was the toxicity test?

The doses were chosen based on both literature data and the results of our previous studies (ref. 16 and ref. 17).

5- How was the extract dissolved and diluted to be given to animals?

The following sentence is added in part 4.2.: Prior to oral administration, the dry extracts were dissolved in distilled water.

6-Statistical analysis is incomplete What is the analysis that was used after the ANOVA to find out the significant differences between the groups and the differences between all groups must be calculated

Thank you for your comments regarding the statistical analysis. After conducting a one-way ANOVA, we utilized the LSD (least significant difference) post hoc test to identify significant differences between the groups. The LSD test is a common and accepted method for pairwise comparisons following ANOVA.

Results

1-All Figures must contain the number of rats that were used for each group with the determination of the statistical analysis ANOVA and after ANOVA and the presence of signs on the columns showing the significant difference between all groups.

Number of Rats in Each Group: The number of rats used in each group has already been detailed in the Materials and Methods section of the article. However, we recognize the value of having this information readily accessible in the figures, and we will add it to the figure descriptions to enhance clarity.

Statistical Analysis Details: The one-way ANOVA test employed in our study is described in the Materials and Methods section. Additionally, we will include information about the LSD post hoc test in that section. To further aid readers, we will also provide a brief mention of the statistical tests used (ANOVA and LSD post hoc test) in the figure descriptions.

Significant Differences Between Groups: Our study primarily focused on comparing all groups with the stress control, as this was most important to our research objectives. We marked the statistical differences by adding * symbol above the columns. While it is also possible to compare individual extracts with combinations and add more symbols, doing so within the figures might lead to complexity and potential confusion due to the numerous comparisons. We could provide detailed information about the significant differences between each pair of groups, including the respective p-values, in supplementary tables. We believe this approach will present the information in an organized and easily digestible format.

Reviewer 2 Report

There are several grammatical errors that have to be corrected. Author must consult a language expert to correct the errors.

Authors have repeated information about cytokines in introduction and discussion. This has to be corrected.

The discussion has to be rewritten. Information not related to the results such as polymorphism and too much details about cytokines should be removed. Instead, the authors should discuss how plant extracts and their phytoconsituents influence anxiety and stress. Further, they should stress how combining herbs may be better than using individual herb or a single phytoconstituent.

The chronic stress model used seems unethical. Stress of different kinds and water deprivation for 24 hours, limited food and water deprivation again are unacceptable.

The use of ether for anesthesia is unacceptable as per international guidelines for animal experiments.

There are several grammatical errors.

Author Response

The authors thank the reviewers for their comments and suggestions, which aim to improve the quality of the manuscript.

Reply to reviewer  2:

There are several grammatical errors that have to be corrected. Author must consult a language expert to correct the errors.

The text has been edited. Corrections are marked in red.

 Authors have repeated information about cytokines in introduction and discussion. This has to be corrected.

 The discussion has to be rewritten. Information not related to the results such as polymorphism and too much details about cytokines should be removed. Instead, the authors should discuss how plant extracts and their phytoconsituents influence anxiety and stress. Further, they should stress how combining herbs may be better than using individual herb or a single phytoconstituent.

Thank you for identifying the repeated information about cytokines in both the Introduction and Discussion sections of our manuscript.

Removed text:

3.1

The proinflammatory cytokine TNF-α is one of the main mediators that induces behavioral changes in rats exposed to chronic stress caused by mild unpredictable stressors. Liu and co-authors who investigated the mechanisms of development of depressive-like symptoms in rats and found that chronic stress leads to increased synthesis of TNF-α in both serum and cerebral cortex. Behavioral changes reported in the forced swimming, tail suspension, and sucrose consumption tests reported depressive-like behavior in animals that correlated with high levels of TNF-α. Increased expression of indoleamine 2,3-dioxygenase, an enzyme that leads to neurotoxicity in the cerebral cortex and decreased serotonin levels, has been observed. These changes correlate with the development of depression in animals [20].

Elevated TNF-α levels in depression are directly related to impaired cognitive processes as a result of decreased brain neutrophil factor in zones CA1 and CA3 in the hippocampus [23].

3.2

There is a polymorphism in the expression of IL-6, which determines individual sensitivity to stress as a factor that leads to the development of depression [42].

Data from a study by Pan et al also show an increase in IL-6 and TNF-α levels in chronically stressed rats, which is associated with the development of depressive-like behavior. Cytokines bind to receptors located on the surface of glial cells and neurons in the hippocampus and hypothalamus and stimulate the hypothalamic-adrenal axis and cortisol secretion. Their long-term effects lead to hormonal imbalance and the development of depression [44].

3.3

IL-10 is one of the most studied cytokines, which plays an important role in the prevention of inflammatory and autoimmune diseases with its immunosuppressive action [53]. Findings from similar studies have yielded mixed results.

We will carefully revise these sections to retain essential information critical to understanding our findings, while avoiding redundancy. In line with your suggestions, we have added a few additional sentences to the Discussion section, accompanied by citations, to explain the effect of phytoconstituents on anxiety, depression, and memory—all of which are related to stress. However, the primary focus of our article remains on the mechanisms by which the extracts influence cytokine levels and the relationship between cytokines and anxiety/depression.We want to point out that we've already published another study focused on the effects of the same extracts on the behavioral aspects of stress. The reference for this study is [17] à Kandilarov, I. K., Zlatanova, H. I., Georgieva-Kotetarova, M. T., Kostadinova, I. I., Katsarova, M. N., Dimitrova, S. Z., Lukanov, L. K., & Sadakov, F. (2018). Antidepressant Effect and Recognition Memory Improvement of Two Novel Plant Extract Combinations - Antistress I and Anti-stress II on Rats Subjected to a Model of Mild Chronic Stress. Folia medica, 60(1), 110–116. https://doi.org/10.1515/folmed-2017-0073.

Regarding the comparison between combining herbs and using individual herbs or a single phytoconstituent, we have already discussed this in Section 3.5 of the Discussion. However, we will review this section to ensure that the points are articulated clearly and will adjust if necessary to emphasize this aspect further.

 The chronic stress model used seems unethical. Stress of different kinds and water deprivation for 24 hours, limited food and water deprivation again are unacceptable.

 The use of ether for anesthesia is unacceptable as per international guidelines for animal experiments.

We are fully aware of the ethical considerations when working with animal models, and we want to assure you that our study was conducted with utmost care for the welfare of the animals. The design of the experiment, including the application of stressors and euthanasia was performed in accordance with accepted guidelines and was approved by the Animal Ethics Committee of the Bulgarian Food Safety Agency with permit No. 127 of 09.12.2015 and the decision of the Ethics Committee at the Medical university of Plovdiv with protocol No. 3 of 21.04.2016. We would like to clarify that the stress method we employed is known as Chronic Unpredictable Mild Stress (CUMS). This model is designed to be one of the mildest forms of stress induction leading to depression and is commonly used in scientific research. Additionally, our approach was chosen over other models such as learned helplessness, maternal separation, restraint stress or olfactory bulbectomy precisely because of its relatively mild nature. à Wang, Qingzhong et al. “The recent progress in animal models of depression.” Progress in neuro-psychopharmacology & biological psychiatry vol. 77 (2017): 99-109. doi:10.1016/j.pnpbp.2017.04.008

We hope these revisions and clarifications address your concerns, and we appreciate your valuable feedback.

Round 2

Reviewer 2 Report

Accept